# Mantis: Interleaved Multi-Image Instruction Tuning

**Dongfu Jiang♣, Xuan He♣◇, Huaye Zeng♣, Cong Wei♣, Max Ku♣, Qian Liu♡, Wenhu Chen♣**
**♣University of Waterloo, ◇Tsinghua University, ♡Sea AI Lab**
`{dongfu.jiang,wenhuchen}@uwaterloo.ca`

**Reviewed on OpenReview:** `https://openreview.net/forum?id=skLtdUVaJa`

## Abstract

Large multimodal models (LMMs) have shown great results in single-image vision language tasks. However, their abilities to solve multi-image visual language tasks is yet to be improved. The existing LMMs like OpenFlamingo, Emu2, and Idefics gain their multi-image ability through pre-training on hundreds of millions of noisy interleaved image-text data from the web, which is neither efficient nor effective. In this paper, we aim to build strong multi-image LMMs via instruction tuning with academic-level resources. Therefore, we meticulously construct MANTIS-INSTRUCT containing 721K multi-image instruction data to train a family of MANTIS models. The instruction tuning empowers MANTIS with different multi-image skills like co-reference, comparison, reasoning, and temporal understanding. We evaluate MANTIS on 8 multi-image benchmarks and 6 single-image benchmarks. MANTIS-Idefics2 can achieve SoTA results on all the multi-image benchmarks and beat the strongest multi-image baseline, Idefics2-8B by an average of 13 absolute points. Notably, Idefics2-8B was pre-trained on 140M interleaved multi-image data, which is 200x larger than MANTIS-INSTRUCT. We observe that MANTIS performs equivalently well on the held-in and held-out benchmarks, which shows its generalization ability. We further evaluate MANTIS on single-image benchmarks and demonstrate that MANTIS also maintains a strong single-image performance on par with CogVLM and Emu2. Our results show that multi-image abilities are not necessarily gained through massive pre-training, instead, they can be gained by low-cost instruction tuning. The training and evaluation of MANTIS has paved the road for future work to improve LMMs' multi-image abilities.

## 1 Introduction

Large Multimodal Models (LMMs) have advanced significantly in recent years. Both closed-source models like GPT-4V (Achiam et al., 2023), Gemini (Team et al., 2023), Reka (Ormazabal et al., 2024), MM1 (McKinzie et al., 2024) and open-source models like LLaVA (Liu et al., 2023c;b), LLaVA-NeXT (Liu et al., 2024), BLIP (Li et al., 2023c), CogVLM (Wang et al., 2023), Idefics (Laurençon et al., 2023) have shown strong visual-language understanding and generation capabilities in single-image tasks like VQA (Antol et al., 2015), TextVQA (Singh et al., 2019b), GQA (Hudson & Manning, 2019), etc. Despite the significant interest in improving LMMs' performance on single-image vision-language tasks, relatively less work attempts to improve LMMs' capability to solve multi-image vision-language tasks. To the best of our knowledge, only a few open-source models (e.g. Idefics (Laurençon et al., 2023), OpenFlamingo (Awadalla et al., 2023), Emu2 (Sun et al., 2023)) and VILA (Lin et al., 2023c) support multi-image inputs.

We argue that multi-image visual ability is also a crucial in real-world applications. Specifically, we categorize the ability into four major skills we want the LMM to encompass: (1) **Co-reference**: understanding the references like "second image" in the natural language expression and grounding it on the referred image to generate a response. (2) **Comparison**: capturing the nuances and commonalities between several images. (3) **Reasoning**: capturing the information across multiple images and reasoning over these multiple pieces

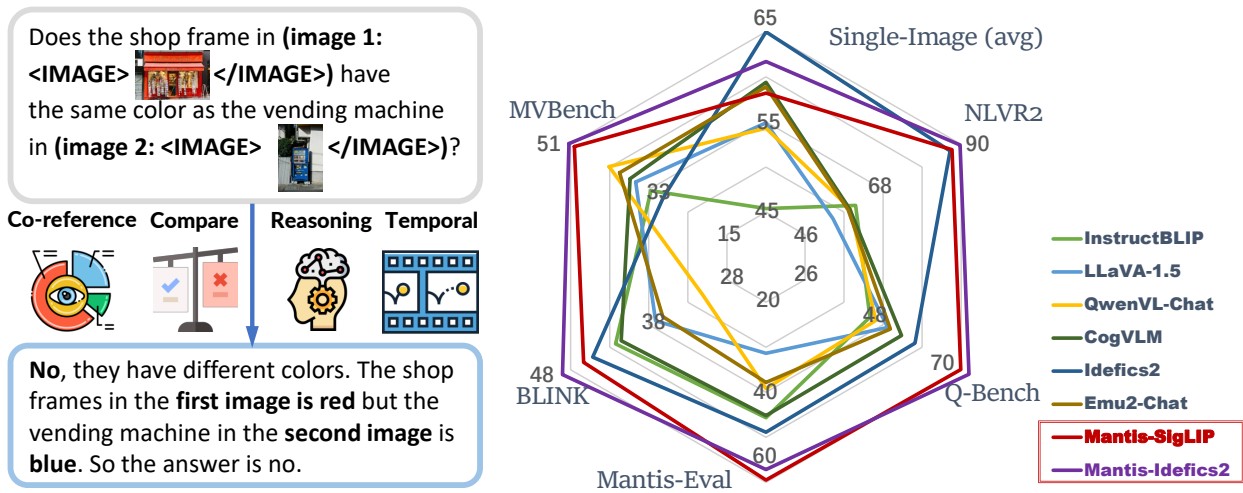

Figure 1: Mantis can accept interleaved text-image to perform different multi-image tasks like coherence, comparison, logical reasoning, and temporal understanding. Mantis achieves state-of-the-art performance on the five multi-image benchmarks and preserves abilities for single-image tasks.

to derive the response. (4) **Temporal understanding**: observing multiple frames of a video or a scene to understand the temporal information like actions, behaviors, interactions, etc.

The existing multi-image LMMs like Idefics (Laurençon et al., 2023), OpenFlamingo (Awadalla et al., 2023), Emu2 (Sun et al., 2023), MM1 (McKinzie et al., 2024), VILA (Lin et al., 2023c) heavily rely on pre-training on a massive interleaved image-text web documents like MMC4 (Zhu et al., 2024), Obelics (Laurençon et al., 2023). For example, MMC4 contains 500 million examples, while Obelics contains 140 million examples. Such a pre-training procedure would require a massive amount of computation, which hampered its adoption in most open-source LMM training pipelines. In this paper, we attempt to build LMMs with academic-level resources to support interleaved multi-image inputs. Specifically, we made a few efforts along this line:

**Dataset**: We create the first multi-image instruction-tuning dataset Mantis-Instruct. It has a total of 721K instances, consisting of 14 subsets to cover all the multi-image skills. Among the 14 subsets, 10 subsets are from the existing datasets. For example, `NLVR2` (Suhr et al., 2018), `IconQA` (Jhamtani & Berg-Kirkpatrick, 2018), etc are used to cover 'reasoning' skill; `DreamSim` (Fu et al., 2023), `Birds-to-Words` (Forbes et al., 2019), etc are used to cover 'comparison' skill; `NExT-QA` (Xiao et al., 2021), `STAR` (Wu & Yu, 2021), etc are used to cover 'temporal understanding' skill. Additionally, 4 new subsets are newly curated, where `LLaVA-665k-multi`, and `LRV-multi` are used to to cover the 'coref' skill, **Contrast-Caption**, and **Multi-VQA** are used to broaden the 'reasoning' skill. The interleaving image placeholders are inserted into the text under various heuristics.

**Architecture**: Similar to LLaVA (Liu et al., 2024), we start from a pre-trained language model like LLaMA-3 (AI@Meta, 2024) or Fuyu (Adept AI, 2023) and a vision transformer encoder from CLIP (Radford et al., 2021) or SigLIP (Zhai et al., 2023). We use the same multimodal projector as LLaVA to map the vision embeddings to the text embeddings space to simply concatenate them with the text embeddings. We also define a text-image interleaving format: "...(image {i}: `<BOI>image embeddings</EOI>`)...", where `<BOI>` and`</EOI>` are the image delimiters. We explored multiple instantiations and found this optimal design.

**Evaluation**: We manually curate a new challenging dataset Mantis-Eval to cover several multi-image skills. This evaluation can help us better analyze LMMs' multi-image abilities.

We train Mantis by combining Mantis-Instruct with another 268K single-image vision-language data to balance the multi-image and single-image abilities. To evaluate the multi-image skills, we adopt 8 benchmarks including NLVR2 (Suhr et al., 2018), BLINK (Fu et al., 2024), MVBench (Li et al., 2023d), Mantis-Eval, etc. to cover all the mentioned skills. We include 15 competitive baseline models including single-image LMMs like LLaVA-Next (Liu et al., 2024), Qwen-VL (Bai et al., 2023) and multi-image LMMs like GPT-4V,

Idefics1/2, etc. For single-image LMMs, we merge multiple images horizontally as a single-image input. By training on 16 x A100 for 36 hours, Mantis achieves state-of-the-art performance on all the multi-image tasks, while preserving competitive performance in single-image tasks. Notably, Mantis-Idefics2 outperforms the strongest baseline Idefics2 (Laurençon et al., 2023) by an average of 13 absolute points on the 7 avaialble benchmarks (excluding MileBench), where Idefics2 is multi-image pre-trained, showcasing the effectiveness of multi-image instruction-tuning. The held-in and held-out results are equivalently strong, which shows the generalization ability of Mantis.

These results are encouraging because we show that the low-cost instruction tuning on 721K high-quality data can lead to a much better generalization performance than other LMMs intensively pre-trained on 100x larger datasets. Mantis also preserves its strong performance of single-image tasks and is on par with CogVLM (Wang et al., 2023) and Emu2 (Sun et al., 2023). We also present insights on the architecture selection and inference approaches based on the results in section 3. We believe Mantis-Instruct can serve as an essential baseline for future studies on multi-image LMMs due to its simplicity and low cost. We will release all the code, data, and models to help with the reproducibility of our results.

## 2 Mantis: Interleaved Multi-Image Instruction Tuning

### 2.1 Method

**Model Architecture:** Most existing LMMs do not support multi-image inputs, either due to the architecture design (Huang et al., 2016; Li et al., 2023c), or the lack of associated code support (Peng et al., 2023; Adept AI, 2023). To enable multi-image training and inference, we have modified the LLaVA architecture and added code support for it. Idefics2 (Laurenccon et al., 2024) already supports multi-image as inputs naturally, we thus inherit their structure.

**Image Context Length:** How many images can a multi-image supported model accept? This is limited by both the LLM backbone's max token length and the number of image tokens the vision encoder generated. For the LLaVA architecture, each image consumes a fixed $(336/14)^2 = 576$ image tokens. For Llama3 with a maximum window size of 8K, we can feed at most 14 images. For Idefics2, the number of tokens per image is resampled to be 64 tokens only due to the existence of perceiver, which is pretty efficient. Given 8K context length, it can accept 128 images at most, even more than some video models.

**Interleaving Text-Image:** A proper text-image interleaving format can help acquire multi-image understanding and reasoning ability. We contend that a good text-image interleaving format should: (1) mark boundaries between images clearly, and (2) denote the serial number of images. Following this principle, we designed our interleaving format as follows: `"(image {i}: <BOI><image><EOI>)"`, where `<BOI>` is the begin of image token and `<EOI>` is the end of image token. `<image>` is the placeholder for image patches. This format adds clear separators between images, and gives serialized information of the image through `"image {i}"`. In practice, we set `<BOI>` and `<EOI>` to be `<Image>` and `</Image>` respectively. Previous work also demonstrates its effectiveness through a comprehensive ablation study (Wu et al., 2024).

### 2.2 Mantis-Instruct: A large scale multiple image question answering dataset

We construct Mantis-Instruct from multiple publicly available datasets. Detailed statistics are shown in Table 1 with several newly curated subset by us. We use a similar dataset format with LLaVA's, where each data item contains multiple images and multiple turns of QA pairs are gathered. We demonstrate some examples of our newly curated subset in Figure 2. We report the number of examples, the number of images per item, the average conversation turns, and the average length. Mantis-Instruct is collected based on the 4 multi-image skills. We here describe the gathered subsets for each skill briefly. Construction details can be found in subsection A.1.

**Co-reference** requires the model to create a mapping from natural language references, such as "second image," to the actual images in the input. It does not require the model to infer across multiple images.

| Subsets | Skill | # Instance | # Avg-I | # Max-I | # Turns | Length$^T$ | Length$^{T+I}$ |
|---------|-------|-----------|---------|---------|---------|-----------|---------------|
| | | Multi-Image Reasoning Datasets | | | | | |
| LLaVA-665k-multi♣ | Coref | 313K | 2.0 | 4 | 21.4 | 558 | 1710 |
| LRV-multi♣ | Coref | 8K | 3.5 | 9 | 83.4 | 2234 | 4251 |
| CoInstruct | Compare | 151K | 2.7 | 4 | 7.5 | 314 | 1620 |
| Dreamsim | Compare | 16K | 3.0 | 3 | 2.0 | 103 | 1831 |
| Spot-the-Diff | Compare | 8K | 2.0 | 2 | 4.0 | 121 | 1273 |
| Birds-to-Words | Compare | 3K | 2.0 | 2 | 2.0 | 101 | 1253 |
| NLVR2 | Reason | 86K | 2.0 | 2 | 2.0 | 105 | 1257 |
| IconQA | Reason | 64K | 2.4 | 6 | 2.0 | 71 | 1454 |
| Contrast-Caption♣ | Reason | 36K | 3.8 | 8 | 7.6 | 871 | 3067 |
| ImageCoDe | Reason | 17K | 10 | 10 | 1.0 | 126 | 5886 |
| Multi-VQA♣ | Reason | 5K | 4.0 | 6 | 11.0 | 1102 | 3417 |
| VIST | Temporal | 7K | 20.3 | 50 | 9.7 | 530 | 12238 |
| NExT-QA | Temporal | 4K | 8.0 | 8 | 17.6 | 572 | 5180 |
| STAR | Temporal | 3K | 8.0 | 8 | 30.2 | 961 | 5569 |
| Total | - | 721K | 4.7 | 50 | 14.4 | 555 | 3584 |
| | | Single-Image Reasoning Datasets | | | | | |
| DVQA | Doc | 200K | 1.0 | 1 | 40.0 | 304 | 880 |
| DocVQA | Doc | 39K | 1.0 | 1 | 2.0 | 62 | 638 |
| ChartQA | Chart | 28K | 1.0 | 1 | 2.0 | 66 | 642 |
| Total | - | 268K | 1.0 | 1 | 14.7 | 144 | 720 |

Table 1: The statistics of MANTIS-INSTRUCT. We list the number of examples, average images per example, maximum image per example, average turns, average text token lengths, and average text+image token lengths. ♣ means the new subsets we constructed in this paper. Here we use Llava's tokenizer and assume each image occupies 576 image tokens, which is the number of tokens for each image. Here "Avg Length$^T$" denotes the average number of tokens for the text part, while "Avg Length$^{T\&I}$" means the average number of tokens of text and images in total. Heuristics about how each subset is curated can be found in subsection A.1.

---

Please generate 10 independent QA pairs. Each question shall involve at least 2 images to answer. Try to cover different ability like reasoning, planning, common sense understanding, etc. Be creative with your questions and **make sure the answers require integration of information from multiple images**. Use "image i" to refer to the i-th image in your questions.

Output format:
Question: First question?
Answer: The answer to the first question.
Question: Second question?
Answer: The answer to the second question.
...

---

Table 2: Propmt template used to curate the `Multi-VQA` subset.

To achieve this, we constructed `LLaVA-665k-multi` and `LRV-multi` by concatenating multiple single-image conversations into a multi-image sequence. Deliberately, we included natural language references like "For the second image" for each question.

**Comparison** requires the model to not only have good co-reference ability but also be able to compare and understand differences across multiple images. We gather `Co-Instruct`, `Dreamsim`, `Spot-the-Diff`, and

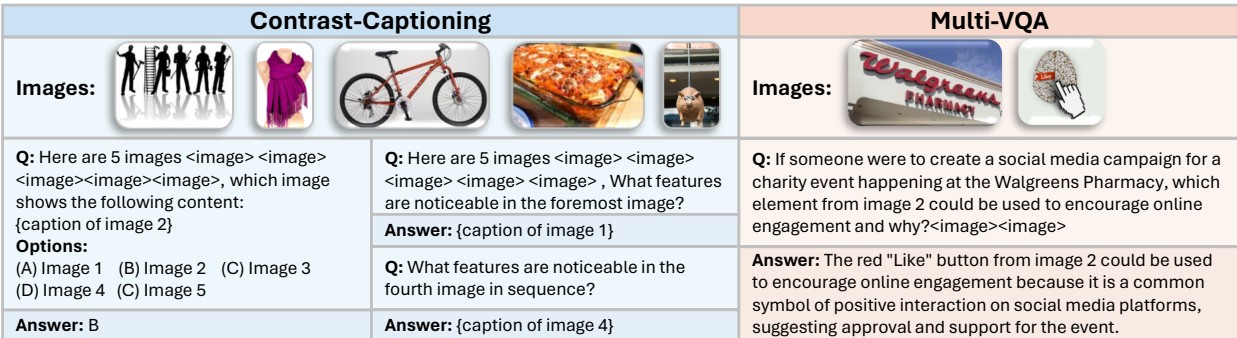

Figure 2: Illustrations of MANTIS-INSTRUCT datasets from Contrast-Captioning and Multi-VQA. Contrastive-Captioning is constructed from existing image captioning datasets to enhance LMM's multi-image 'co-reference' ability. Multi-VQA is GPT-4V synthesized multi-turn multi-image data used to enhance LMM's multi-image 'reason' ability.

`Birds-to-Words` together to enhance the model's comparison ability. They cover a wide range of topics such as image quality, visual similarity, and difference description.

**Reasoning** requires the model to further make inferences by combining its world knowledge with the information collected using the co-reference and comparison ability. It distinguishes from the comparison skill by introducing more complex human instruction associated with the images. We gather `NLVR2`, `IconQA`, `Contrast-Caption` by reformatting captioning datasets, `ImageCoDe`, and our self-collected `Multi-VQA` to further equip the model with reasoning ability. The topics include logical reasoning, counting, image matching, image retrieval, and free-form multi-image QA. `Multi-VQA` is synthesized by prompting GPT-4V, where the prompt template is shown in Table 2.

**Temporal Understanding** requires the model to understand temporal image sequences such as videos, comics, etc. We include a small number of video understanding datasets (14K), `VIST`, `NExT-QA`, and `STAR`, to activate the model's temporal understanding ability. We contend that a model with the above 3 multi-image skills shall be easily tuned to understand image sequences.

## 2.3 Heuristics of data curation

Similar to visual instruction tuning (Liu et al., 2023c), data curation is an important step to ensure good performance. During the curation, we found some heuristics that could improve performance. We list them below. Note that the first heuristics is a simple way to diversify QA format to avoid overfitting, and the rest two heuristics have already been proven effective by previous work Co-Instruct (Wu et al., 2024). We thus adopt them directly to avoid duplicate efforts to confirm their effectiveness.

1. **Conflicts between short answers and long answers.** Many datasets we collected are domain-specific, and some of them are classification tasks like NLVR2 (Suhr et al., 2018) and DreamSim (Fu et al., 2023). For these datasets, we convert them into multiple-choice QA format to avoid conflicts between repeated short-answers and some long answers from other datasets.

2. **Addition of image denotation in the text for each question.** We manually add image denotations like "the first image", "image 2", "the right image", etc. to each question in our dataset to make sure the question involved with multiple images is clear to answer.

3. **Positions of the "<image>" placeholder.** For each item with multiple images, we write some rules to randomly put the image placeholders at the beginning of the first question or the end of the first question. This technique is also used by the curation LLaVA-665k (Liu et al., 2023b).

# 3   Experiments

In this section, we aim to evaluate MANTIS's ability on both multi-image and single-image vision-language tasks. For multi-image tasks, depending on whether the LMM can handle multiple images, we either concatenate multiple images ('merge') horizontally as a single image or feed multiple images as a 'sequence' to the model. We have trained four model variants listed in Table 3, where we list the architecture and dataset details. Mantis-CLIP and Mantis-SigLIP only adopt the CC3M 560K subset for feature alignment. Mantis-Flamingo and Mantis-Idefics2 are initialized from OpenFlamingo and Idefics2, respectively, which have been intensively pre-trained on the multi-image corpus. We also include Mantis-Fuyu to investigate the significance of vision encoder in the LMM architecture. After this pre-training and preparation, we fine-tune them on MANTIS-INSTRUCT to get Mantis.

| Mantis-Models | V-Encoder | LLM-backbone | Pre-Training (PT) | PT size | Fine-Tuning | # Tokens / I |
|---|---|---|---|---|---|---|
| Mantis-Fuyu | - | Persimmon-8B | Unknown | Unknown | Mantis-Instruct | [1,2304] |
| Mantis-Flamingo | CLIP | MPT-7B | MMC4, LAION | 2421M | Mantis-Instruct | 576 |
| Mantis-CLIP | CLIP | LLaMA-3-8B | CC3M subset | 0.56M | Mantis-Instruct | 576 |
| Mantis-SIGLIP | SigLIP | LLaMA-3-8B | CC3M subset | 0.56M | Mantis-Instruct | 576 |
| Mantis-Idefics2 | SiGLIP | Mistral-7B-v0.1 | OBELICS, Cauldron | 143M | Mantis-Instruct | 64 |

Table 3: Architecture and training details of mantis model family. "I" for "Image" in last column.

## 3.1   Training Details

For MANTIS-CLIP and MANTIS-SigLIP, we first pre-train the multimodal projector on LLaVA pre-train data, where the learning rate is set to 1e-3, the batch size is set to 256. After pre-training, we fine-tune the 2 models on the MANTIS-INSTRUCT. For MANTIS-Fuyu, MANTIS-Flamingo and MANTIS-Idefics2, we directly fine-tune them on MANTIS-INSTRUCT since they have already been pre-trained on millions of data.

During the fine-tuning, we train each model on the data for 1 epoch, with a batch size of 128. The maximum context length is set to 8192. The learning rate is all set to 1e-5, except for Idefics2, where the learning rate is set to 5e-6 to better preserve its original knowledge. We set the warmup ratio to be 0.03 and use a cosine learning rate scheduler. We speed up our training and inference with Flash-attention2 (Dao, 2023). We use DeepSpeed Zero-3 (Aminabadi et al., 2022) for full fine-tuning. We apply QLoRA (Dettmers et al., 2023) along with DoRA (yang Liu et al., 2024) to more efficiently do comprehensive ablation studies under limited resources. All full fine-tuning ran on 16 A100 GPUs while the ablation study ran on 8 A100 GPUs.

## 3.2   Baselines

For 'merge' evaluation, we include BLIP-2-13B (Li et al., 2023c), InstructBLIP-13B (Dai et al., 2023), Qwen-VL-Chat (Bai et al., 2023), LLaVA-1.5-7B (Liu et al., 2023c), LLaVA-1.6-7B (Liu et al., 2024), Kosmos2 (Peng et al., 2023), and Fuyu-8B (Adept AI, 2023). For 'sequence' evaluation, we include OpenFlamingo (Awadalla et al., 2023), Otter (Li et al., 2023b), VideoLLaVA (Lin et al., 2023a), Emu2-Chat-34B (Sun et al., 2023), VILA (Lin et al., 2023b), Idefics1 (Laurençon et al., 2023), Idefics2 (Laurençon et al., 2023), GPT-4V (Achiam et al., 2023). Additionally, for single-image tasks, we include InstructBLIP-7B-Vicuna (Dai et al., 2023), Yi-VL-6B (AI et al., 2024).

## 3.3   Evaluation Benchmarks

For multi-image tasks, we use 2 held-in benchmarks: NLVR2 (Suhr et al., 2018) and Qbench (Wu et al., 2023a); and 3 held-out benchmarks: Mantis-Eval, BLINK (Fu et al., 2024), and MVBench (Li et al., 2023d). We report detailed statistics in Table 4.
**NLVR2 (Suhr et al., 2018)** evaluates the model's ability to conduct logical reasoning across the contents of images. It asks the model to compare the contents of the two given images and judge whether a given statement is correct or not. All questions are multiple-choice. We use the test-public split for evaluation.

| Benchmarks | Skill | Type | # Instances | # Avg-I | # Max-I | Lenght$^{T+I}$ |
|---|---|---|---|---|---|---|
| NLVR2$^\dagger$ | Reason | MCQ | 6,967 | 2.0 | 2 | 1,190 |
| Q-Bench$^\dagger$ | Comparison | MCQ | 1,000 | 2.0 | 2 | 1,167 |
| Mantis-Eval | Reason & Coref | MCQ & Short | 217 | 2.5 | 5 | 1,455 |
| BLINK | Reason | MCQ | 1,901 | 1.9 | 4 | 1,129 |
| MVBench | Temporal | MCQ | 4,000 | 8.0 | 8 | 4,625 |
| MileBench | Mixed | MCQ | 6,440 | 15.2 | 109 | 9,294 |
| MuirBench | Mixed | MCQ | 2,600 | 4.3 | 9 | 2,502 |
| MMIU | Mixed | MCQ | 11,600 | 6.6 | 32 | 3,852 |

Table 4: Statistics of the evaluation dataset. Held-in benchmarks, NLVR2 and Q-Bench, are marked with $\dagger$ notation. The rest of them are held-out benchmarks. "Mixed" means a .

**Qbench (Wu et al., 2023a)** is a benchmark evaluating whether LMMs can properly judge and compare the quality of a benchmark. Q-bench aims to evaluate the low-level visual abilities of LMMs, such as judging the image quality. All questions are multiple-choice. In our experiments, we evaluate the Qbench2-A2-pair dev set, where a low-level visual question is asked based on multiple image contents.

**Mantis-Eval** consists of 217 multiple-image reasoning examples that cover different topics, including size perceptions, weight comparisons, etc. This dataset is carefully curated by our annotators, where images are acquired via Google Search, and then come up with a proper question which requires understanding the contents in the two images well to answer. Mantis-Eval contains both multiple-choice and short-answer questions. We report results in the test split.

**BLINK (Fu et al., 2024)** is a benchmark on core visual perception abilities, where humans can solve most tasks "within a blink" (e.g., relative depth estimation, visual correspondence, forensics detection, and multi-view reasoning). Some questions in the benchmark involved multiple images, such as image similarity comparison. We report results in the validation set of the benchmark.

**MVBench (Li et al., 2023d)** is a comprehensive multimodal video understanding benchmark covering 20 challenging video tasks. The questions cannot be solved with a single frame, necessitating understanding an image sequence to solve the questions in the benchmark. We extract 8 frames per video for the evaluation. We report results in test split.

**MileBench (Song et al., 2024)** is a novel benchmark designed to test the multimodal long-context capabilities of MLLMs. It comprises realistic evaluation and diagnostic evaluation. The former involves tasks like temporal understanding and semantic multi-image comprehension. The later diagnostic evaluation involves TextNeedle and ImageNeedle retrieval tasks. We only reported the performance of the realistic evaluation. The results of the diagnostic evaluation can be found in MileBench's paper.

**MuirBench (Wang et al., 2024a)** is a comprehensive benchmark consisting of 12 diverse multi-image tasks, such as scene understanding, ordering, etc. It contains 2,600 multiple-choice questions with 11,264 images involved in total. We report the overall average performance across the 12 tasks.

**MMIU (Meng et al., 2024)** is a multi-image evaluation benchmark encompassing 7 types of multi-image relationships, 52 tasks, 77K images, and 11K multiple-choice questions. We report the overall average performance across all the tasks.

### 3.4 Results on Multi-Image Tasks

We report our results on the multi-image task in Table 5. We put the results of GPT-4V/o at the top of the table as it's a closed-source LMM. We can see that our best version Mantis-Idefics2 almost matches the overall performance of GPT-4V. We put comparison results between MANTIS and other open-source LMMs, along with the insights derived in the following paragraphs.

**Mantis performs well on held-in evaluation**: We found that Mantis-SigLIP without any multi-image pre-training can already achieve the best-known performance on these two benchmarks. Mantis-Idefics2 has better performance of 89.71 and 75.20 on NLVR2 and Q-Bench respectively, marking it the SoTA for these two tasks. Notably, Mantis-Idefics2 also beats the Idefics2 by 2.84 points, which was already trained on NLVR2. It shows the cross-task benefits of our instruction tuning. The remarkable performance on these 2

| Models | Size | NLVR2 | Q-Bench | Mantis-Eval | BLINK | MVBench | MileBench | MuirBench | MMIU |
|---|---|---|---|---|---|---|---|---|---|
| GPT-4V/o | - | 88.80 | 76.52 | 62.67 | 51.14 | 43.50 | 53.00 | 68.00 | 55.70 |
| *Open Source Models (merge)* | | | | | | | | | |
| Random | - | 48.93 | 40.20 | 23.04 | 38.09 | 27.30 | 22.30 | 23.99 | 27.40 |
| Kosmos2 | 1.6B | 49.00 | 35.10 | 30.41 | 37.50 | 21.62 | - | - | - |
| LLaVA-v1.5 | 7B | 53.88 | 49.32 | 31.34 | 37.13 | 36.00 | 38.00 | 23.46 | 19.20 |
| LLaVA-V1.6 | 7B | 58.88 | 54.80 | 45.62 | 39.55 | 40.90 | 38.10 | - | 22.20 |
| Qwen-VL-Chat | 7B | 58.72 | 45.90 | 39.17 | 31.17 | 42.15 | 39.10 | - | 15.90 |
| Fuyu | 8B | 51.10 | 49.15 | 27.19 | 36.59 | 30.20 | - | - | - |
| BLIP-2 | 13B | 59.42 | 51.20 | 49.77 | 39.45 | 31.40 | - | - | - |
| InstructBLIP | 13B | 60.26 | 44.30 | 45.62 | 42.24 | 32.50 | - | - | - |
| CogVLM | 17B | 58.58 | 53.20 | 45.16 | 41.54 | 37.30 | - | 20.85 | - |
| *Open Source Models (sequence)* | | | | | | | | | |
| OpenFlamingo | 9B | 36.41 | 19.60 | 12.44 | 39.18 | 7.90 | 27.40 | 23.73 | - |
| Otter-Image | 9B | 49.15 | 17.50 | 14.29 | 36.26 | 15.30 | - | - | - |
| Idefics1 | 9B | 54.63 | 30.60 | 28.11 | 24.69 | 26.42 | - | 35.43 | 12.80 |
| VideoLLaVA | 7B | 56.48 | 45.70 | 35.94 | 38.92 | 44.30 | - | - | - |
| Emu2-Chat | 37B | 58.16 | 50.05 | 37.79 | 36.20 | 39.72 | - | - | - |
| VILA-1.5 | 8B | 76.45 | 45.70 | 51.15 | 39.30 | 49.40 | 44.40 | - | - |
| Idefics2 | 8B | 86.87 | 57.00 | 48.85 | 45.18 | 29.68 | - | 26.08 | 27.80 |
| Mantis-Fuyu | 8B | 72.41 | 54.10 | 37.79 | 39.87 | 39.12 | - | - | - |
| Mantis-Flamingo | 9B | 52.96 | 46.80 | 32.72 | 38.00 | 40.83 | - | - | - |
| Mantis-CLIP | 8B | 84.66 | 66.00 | 55.76 | 47.06 | 48.30 | - | 37.38 | - |
| Mantis-SIGLIP | 8B | 87.43 | 69.90 | **59.45** | 46.35 | 50.15 | **47.5** | 36.12 | - |
| Mantis-Idefics2 | 8B | **89.71** | **75.20** | 57.14 | **49.05** | **51.38** | - | **44.50** | **45.60** |
| Δ over SOTA | - | 2.84 | 18.20 | 8.30 | 3.87 | 1.98 | 3.10 | 9.07 | 17.80 |

Table 5: Evaluation results on multi-image benchmarks. The highest score of open-source models for each task is bolded. The highest score of the open-source baseline model is underscored. Δ is the performance gains of MANTIS compared to the best baseline model for each task. We report the GPT-4o's results on MuirBench and MMIU, and GPT-4V's on the rest of them.

| Model | Size | TextVQA | VQA | MMB | MMMU | OKVQA | SQA | Avg |
|---|---|---|---|---|---|---|---|---|
| OpenFlamingo | 9B | 46.3 | 58.0 | 32.4 | 28.7 | 51.4 | 45.7 | 43.8 |
| Idefics1 | 9B | 39.3 | 68.8 | 45.3 | 32.5 | 50.4 | 51.6 | 48.0 |
| InstructBLIP | 7B | 33.6 | 75.2 | 38.3 | 30.6 | 45.2 | 70.6 | 48.9 |
| Yi-VL | 6B | 44.8 | 72.5 | 68.4 | 39.1 | 51.3 | 71.7 | 58.0 |
| Qwen-VL-Chat | 7B | 63.8 | 78.2 | 61.8 | 35.9 | 56.6 | 68.2 | 60.8 |
| LLaVA-1.5-Vicuna | 7B | 58.2 | 76.6 | 64.8 | 35.3 | 53.4 | 70.4 | 59.8 |
| LLaVA-1.6-Vicuna | 7B | 64.9 | 81.8 | 67.4 | 35.8 | 44.0 | 68.5 | 60.4 |
| Emu2-Chat | 37B | 66.6 | **84.9** | 63.6 | 36.3 | **64.8** | 65.3 | 63.6 |
| CogVLM | 17B | **70.4** | 82.3 | 65.8 | 32.1 | 64.8 | 65.6 | 63.5 |
| VILA-1.5 | 8B | 64.4 | 80.9 | 72.3 | 36.9 | 0.0 | 79.9 | 55.7 |
| Idefics2 | 8B | 70.4 | 79.1 | 75.7 | **43.0** | 53.5 | **86.5** | **68.0** |
| Mantis-CLIP | 8B | 56.4 | 73.0 | 66.0 | 38.1 | 53.0 | 73.8 | 60.1 |
| Mantis-SigLIP | 8B | 59.2 | 74.9 | 68.7 | 40.1 | 55.4 | 74.9 | 62.2 |
| Mantis-Idefics2 | 8B | 63.5 | 77.6 | **75.7** | 41.1 | 52.6 | 81.3 | 65.3 |

Table 6: Evaluation results of MANTIS on various single-image tasks. The full name of the abbreviations in the table are listed here. TextVQA: TextVQA validation set; VQA: VQA-v2 validation set; MMB: MMBench-English test set; SQA: ScienceQA-Image validation set. The highest score for each task is bolded, while the second highest is underscored.

benchmarks demonstrates that Mantis has gained strong ability in multi-image logical reasoning and low-level vision understanding.

**Mantis generalizes well on held-out evaluation**: We show that our models can attain strong performance on the 6 held-out benchmarks. Mantis-SigLIP achieves 59.45 on Mantis-Eval, and Mantis-Idefics2 achieves 49.05, and 51.38 on BLINK and MVBench respectively, surpassing all other baselines on these held-out tasks. It's also worth mentioning the reported performance of Mantis on the 3 recently released multi-image benchmarks with comprehensive evaluation tasks. Mantis models surpassed the VILA and Idefics2, which are pre-trained with millions of multi-image data, by a large margin, making them the best open-source models on the leaderboard. This is strong evidence of Mantis's good generalization ability in the unseen scenario.

**Multi-Image Pre-training is not necessary**: In these experiments, Mantis-CLIP and Mantis-SigLIP are attaining much better performance than Mantis-Flamingo, and similar performance as Mantis-Idefics, which are pre-trained on millions of text-image interleaved data. These results reveal that the multi-image pre-training (on massive web corpus) is not a necessary path toward strong multi-image performance.

**'Merge' vs 'Sequence' input**: The 'merge' evaluation is an important baseline to evaluate the multi-image understanding ability of those LMMs that can only accept one image at a time. LLaVA-v1.6's dynamic image resolution feature can divide a single image into multiple tiles, processing the 'merge' image like multiple images (Liu et al., 2024). But it still performs poorly on multi-image tasks, only getting 45.62 and 39.55 on the Mantis-Eval and BLINK benchmark, which are at least 10 absolute points lower than Mantis. This result indicates the advantages of building models to accept sequence image inputs.

**Vision encoder matters**: Unlike many other LMMs like Flamingo, LLaVA, and BLIP, Fuyu is a decoder-only LMM without a vision encoder. To investigate the importance of vision encoder, we also train an additional Mantis-Fuyu as a comparison. Results in Table 5 show that the performance of Mantis-Fuyu is worse than Mantis-CLIP and Mantis-SigLIP where a vision encoder exists. While we acknowledge a possible factor brought by the different LLM backbones, we think whether vision encoder exists is a bigger difference and conclude that a good vision encoder can enhances the performance.

### 3.5   Results on Single-Image Tasks

We also evaluate Mantis-CLIP and Mantis-SigLIP on various single-image tasks, including TextVQA (Singh et al., 2019a), VQA-v2 (Goyal et al., 2016), MMBench (Liu et al., 2023d), MMMU (Yue et al., 2023), etc. Results are shown in Table 6. All the evaluations are conducted with the help of LMMs-Eval (Li* et al., 2024) tool. Mantis-Flamingo's and Mantis-Fuyu's performance is omitted due to its bad performance.

Compared to a set of popular LMMs, Mantis-SigLIP gets significant improvements on MMBench-English, MMMU, and ScienceQA benchmarks though we did not specifically optimize the single-image abilities. Solving problems in these tasks requires not only the visual perception ability but also the good reasoning ability of the LLM backbone. Considering that we are using the powerful LLaMA-3 as the backbone LLM, these improvements are expected. We also compare the results of Mantis-Idefics2 against Idefics2 on single-image tasks. On several tasks like MMB, MMMU, and OKVQA, Mantis-Idefics maintains a similarly strong performance as Idefics2. However, we do observe some drops in ScienceQA, which are mainly due to a lower mix ratio of OCR and Math data in our instruction tuning. Overall, the average drop of 4% is tolerable.

### 3.6   Ablation Studies

**Is multi-image instruction tuning necessary?** To investigate, we further train a Mantis-Flamingo based on OpenFlamingo (Awadalla et al., 2023), which has been trained on MMC4 (Zhu et al., 2023), a text-image interleaved pre-training dataset. As shown in Table 5, the average performance of Mantis-Flamingo increased by 19.2 points after further fine-tuning on MANTIS-INSTRUCT. The results show that there is still a huge improvement space, even if a model is fully pre-trained on a multi-image dataset, which demonstrates the necessity of learning multi-image skills in the fine-tuning phase.

**Is multi-image pre-training necessary?** To investigate, we design an ablation study, where one model is pre-trained on the CC3M 560K subset, which contains only image-text pairs, while the other one is pre-trained additionally on OBELICS-100K subset, which contains interleaved multi-image-text pairs. Due

| Pretrain | NLVR2 | Q-Bench | Mantis-Eval | BLINK | MVBench | Avg |
|---|---|---|---|---|---|---|
| CC3M subset + MantisInstruct subset | 76.73 | 57.00 | 47.00 | 43.79 | 47.08 | 54.32 |
| CC3M subset + MantisInstruct subset +OBELICS-100K | 76.83 | 60.50 | 53.00 | 45.20 | 45.97 | 56.30 |
| Δ (with OBELICS-100K) | +0.10 | +3.50 | +6.00 | +1.41 | -1.11 | +1.98 |

Table 7: Ablation on whether multi-image pre-training is necessary. One model is pre-trained on the CC3M subset, while the other is additionally pre-trained on the a OBELICS-100K multi-image dataset. They are both fine-tuned on a subset MANTIS-INSTRUCT for comparison. Results show minor improvement with additional multi-image pre-training. Each ablation model is trained with QLoRA with DoRA.

| Models | NLVR2 | Q-Bench | Mantis-Eval | BLINK | MVBench | Avg | Δ |
|---|---|---|---|---|---|---|---|
| Idefics2-8B | 86.87 | 57.00 | 48.85 | 45.18 | 29.68 | 53.52 | - |
| +Coref | 84.48 | 63.20 | 57.14 | 48.58 | 49.40 | 60.56 | +7.04 |
| +Compare | 85.92 | 69.00 | 53.92 | 48.00 | 48.33 | 61.03 | +0.47 |
| +Reason | **89.62** | **68.50** | **62.21** | 48.31 | 47.93 | 63.31 | +2.28 |
| +Temporal | 89.54 | 67.00 | 61.29 | **49.34** | **50.02** | **63.44** | +0.12 |

Table 8: Data ablation study results of different subsets of MANTIS-INSTRUCT, where each skill corresponding to a specific subsets defined in Table 1. Δ column of each row represents the performance improvements after adding new subsets to the last column results. Each ablation model is trained with QLoRA with DoRA.

to limited computation resources, we fine-tune both models on a downsampled MANTIS-INSTRUCT dataset to compare their performance. Results in Table 7 show that there are little improvements by adding additional pre-training on OBELICS-100K. We thus conclude that multi-image pre-training could be useful, but it is not a necessary path for models to gain multi-image abilities. The poor performance on Mantis-Flamingo also confirm our hypothesis.

**Ablation study of Mantis-Instruct components:** We conduct a data ablation study to analyze the effects of different subsets in MANTIS-INSTRUCT to enhance a model's multi-image ability. As shown in Table 8, As new subsets of MANTIS-INSTRUCT are added continually, the overall performance is continually improving, proving that every subset is contributing positively.

### 3.7 Qualitative Results

To conduct a qualitative analysis of Mantis models on specific multi-image scenarios, we include some case studies in Figure 3, where we include Emu2, which can accept a sequence of images as inputs, and LLaVA-1.5, which can only accept one image at a time, and the powerful GPT-4V, which perform well on both single-image and multi-image scenario.

The first case requires the model to count the number of dice respectively in the two images. While previous LMMs might have learned good counting ability in the single-image scenario, most of them failed in the multi-image counting problem, including GPT-4V. The second case requires the model to involve their world knowledge to infer the mood of the character in three images. However, Emu2, and LLaVA-1.5 output similar contents, and both fall back into the simple captioning behavior. It demonstrates that these models struggle to distinguish two images as separate information. In contrast, Mantis can correctly do the task due to the existing of reasoning subset in Mantis-Instruct.

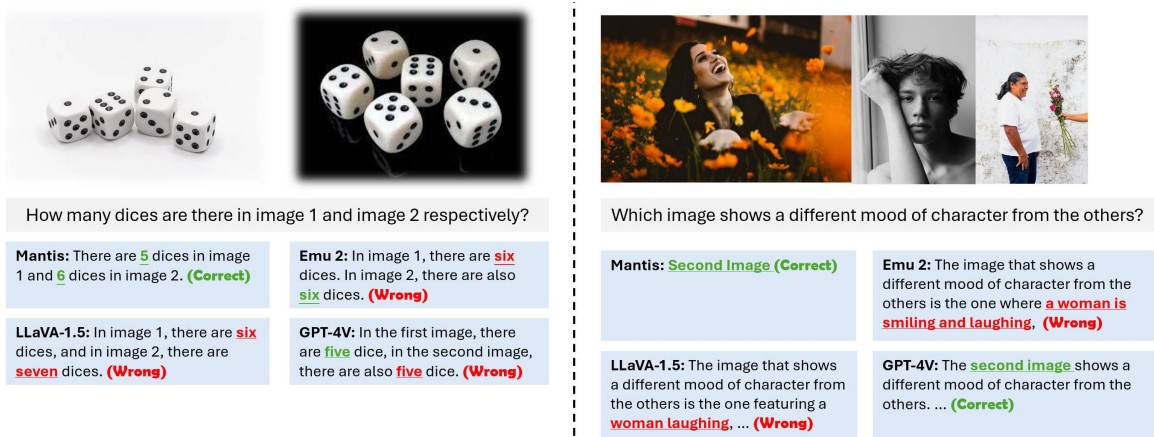

Figure 3: Two case studies in multi-image scenario. MANTIS can understand multiple images and complete the given task, thanks to the 4 multi-image skills learned from MANTIS-INSTRUCT.

# 4 Related Works

## 4.1 Large Multimodal Models

Large Multimodal Models (LMMs) are mostly denoted as large language models that can understand multiple modalities besides the human language. Some works aim to fuse image, audio, video, or more modalities with language (Wu et al., 2023b; Zhan et al., 2024), while more works mainly focus on better-combining vision knowledge with the language (Alayrac et al., 2022; Awadalla et al., 2023; Dai et al., 2023). BLIP-2 has curated a large-scale image captioning dataset, bootstrapping a language model along with a vision encoder to be a powerful multimodal model (Li et al., 2023c). Following this, LLaVA further proposes a cheaper approach to train a powerful LMM through visual instruction tuning (Huang et al., 2016). LLaVA-Next further optimizes single-image performance, with a cost of an increasing number of image tokens per token (Liu et al., 2024), consuming more than 2k for each image, which is about 4 times of the original LLaVA model. Later LMMs like QwenVL (Bai et al., 2023), CogVLM (Wang et al., 2023), Yi-VL (AI et al., 2024), etc. all follow a similar architecture of LLaVA. However, real-world visual problems are way more complex and usually require a sequence of images to describe the situation. Although better performance is achieved in the single-image scenario, it's doubtful whether it's worth the cost.

## 4.2 Multi-Image Supported Large Multimodal Models

Previous works have also noticed the importance of multi-image ability. Deepmind's close-sourced Flamingo (Alayrac et al., 2022) focuses on optimizing in-context learning for a sequence of image/text pairs but lacking free-from image-text interleaved training examples. Kosmos2 (Peng et al., 2023), Fuyu (Adept AI, 2023) both support text-image interleaved format given the model structure, but they did not optimize in multi-image reasoning, and the released inference codes also do not support multi-image as input. Emu2 (Sun et al., 2023) is a generative multimodal model that supports interleaved text-image inputs, as well as generating both images and texts. However, it's mainly focused on in-context examples like Flamingo and auto-regressive image generation, instead of the 4 skills mentioned in section 1 Another common variant of LMMs that can accept multiple images as input is video understanding models, such as Video-LLaMA (Zhang et al., 2023a). However, as shown in the MVBench benchmark, it is also shown that Video-LLaMA behaves worse than LMMs that can only accept one image as input (Li et al., 2023d), which raises doubts about its multi-image understanding ability. Different from these works, Mantis aims to optimize the multi-image ability of LMMs guided by the 4 defined skills, co-reference, reasoning, comparison, and temporal understanding, thus equipping the model with better multi-image ability. Our contemporary works on optimizing interleaved multi-image ability include LLaVA-Next-Interleave (Li et al., 2024b) and LLaVA-OneVision (Li et al., 2024a), which also collects large scale of single-image, multi-image, and video understanding data to train a more

powerful vision language model. These two works appear after our work and incorporate our Mantis-Instruct as their training data, demonstrating our dataset's value.

### 4.3 Multi-Image Evaluation

While many benchmarks have been developed for single-image tasks, such as TextVQA (Singh et al., 2019a), VQA-V2 (Goyal et al., 2016), MMBench (Liu et al., 2023d), and so on, the multi-image evaluation did not capture much attention until recently. Difference description is the first task which large-scale data and good evaluation benchmarks are developed for, including works like Spot-the-Diff (Jhamtani & Berg-Kirkpatrick, 2018), Birds-to-Words (Forbes et al., 2019), etc. They are usually evaluated by n-gram metrics like BLEU, and ROUGE, and only involve two images. The later NLVR2 (Suhr et al., 2018) further includes logical reasoning across 2 images into evaluation, focusing on a higher level of cognition ability. Q-bench (Wu et al., 2023a) and BLINK (Fu et al., 2024) are both proposed for low-level multi-image vision tasks, where humans easily do but model struggles. Another set of multi-image evaluations comes from video or image sequence evaluation, including Mementos (Wang et al., 2024b), MVBench (Li et al., 2023d), etc. In this case, the model usually needs to take more than 2 images to answer. Recently, there are also attempts to prompt LMMs to evaluate vision-language task (Zhang et al., 2023b), or image-generation tasks (Ku et al., 2023), which also demands good multi-image reasoning ability. The accuracies of the these tasks of different LMMs can be seen as another form of benchmark for multi-image ability.

## 5 Conclusion

We propose Mantis, a new large multimodal model family designed to be able to accept interleaved text-image inputs. Mantis is fine-tuned on our Mantis-Instruct, a text-image interleaved dataset consisting of 721K examples, covering 4 crucial skills, including co-reference, reasoning, comparison, and temporal understanding, for multi-image tasks. We also propose Mantis-Eval, a high-quality multi-image evaluation benchmark consisting of 217 examples with an average of 2.5 images per example. Comprehensive experiments show that Mantis models effectively acquire the 4 multi-image skills and achieves state-of-the-art performance on 5 multi-image evaluation benchmarks, and preserve decent single-image reasoning performance. We shared insights about the model architecture and inference approaches based on the experiment results. Extending the image sequence context length and compressing the number of tokens per tokens efficiently are regarded as our future work.

### Limitations

Our work mainly focuses on improving the multi-image ability of LMMs. Our collected dataset Mantis-Instruct reformat some existing multi-image datasets, which might still contain some noise due to the quality of the original collection. Besides, we found that most of the examples in the Mantis-Instrut tend to be multiple-choice QA questions, where the response tends to be short, resulting in the current version of Mantis models tending to output shorter responses instead of long reasoning texts and falling short in instruction-following ability in the multi-image scenario. Although we have introduced `Multi-VQA` subset to alleviate this issue, this problem still exists, and we will leave it to future work by introducing more long-form multi-image data. While enhancing the multi-image ability well, we find out that there is a trend of performance degeneration in the single-image performance compared to the original model Table 6. Some single-image QA datasets have been introduced to alleviate this issue, but the conflicts between multi-image and single-image ability seem to exist continually. How to balance these two abilities will also be our future direction.

### Societal Impacts

Our proposed model, Mantis, and released dataset, Mantis-Instruct, can have substantial impacts on society through the application of Large Multimodal Models. LMMs with multi-image ability can analyze, and reason in real-world scenarios, helping machines, and robots, to make decisions automatically, assisting people in website browsing, travel planning based on multiple maps and pictures, etc. We believe Mantis can serve

as one of the first useful open-source LMMs that make progress in these applications to make these scenarios possible and efficient.

However, there is a potential that our Mantis can generate hallucinations, make the wrong decisions, and fail to reason across complex real-world scenarios. While we have tried to improve Mantis's accuracy and performance in instruction-following and multi-image understanding by carefully training on high-quality Mantis-Instruct, Mantis can still sometimes make mistakes, thus harming its faithfulness as a useful LMM tool. There is also the potential for misuse of our model and dataset after we make them open source. Though we have added proper licenses and terms of usage to our assets, misuse cannot be safely eliminated.

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

# A    Appendix / supplemental material

## A.1    Details of Mantis-Instruct subsets

We construct MANTIS-INSTRUCT from multiple publicly available datasets. Detailed statistics are shown in Table 1. We use a similar dataset format with LLaVA's, where each data item contains multiple images and multiple turns of QA pairs are gathered. We report the number of examples, the number of images per item, the average conversation turns, and the average length. We describe the task type of each dataset and the processing techniques used in the following:

**LLaVA-665k-multi**: Besides the above-mentioned multi-image reasoning tasks, we also manually reformat the LLaVA-655k dataset from LLaVA-1.5 into a "synthetic" multi-image dataset. Since each original data item consists of multiple QA pairs about a single image, we randomly merge 2 to 4 data items into a multi-image data item. Then we add proper image denotation like "For the second image, ..." for each question, and shuffle all the QA pairs to form the final multi-image data item. Answering each question in this "synthetic" dataset does not require reasoning across images, but demands high image co-reference ability. Besides, it also helps avoid forgetting single-image ability in multi-image scenarios and increases instruction-following ability.

**Co-Instruct (Wu et al., 2024)**: Co-Instruct is a dataset curated to compare the quality of 2-4 images in open-ended answer or multi-choice QA format. It's similar to the tasks in the Qbench (Wu et al., 2023a).

**NLVR2 (Suhr et al., 2018)**: NLVR2 is a natural language reasoning dataset grounded in images. The task is to determine whether a statement is true or false given the visual contents of 2 images. NLVR2 covers many linguistic phenomena, such as cardinality, existential quantifiers, universal quantifiers, etc.

**Dreamsim (Fu et al., 2023)**: DreamSim proposes a dataset to train a model to judge image similarity. We reuse their datasets, and the task is to judge which candidate image is more similar to a reference image.

**ImageCoDe (Krojer et al., 2022)**: ImageCoDe is a multimodal challenge called image retrieval from contextual descriptions, where a multimodal model is required to retrieve/select the correct image from a set of 10 minimally contrastive images. For each image, 9 similar images are retrieved from Open Images Dataset V6 (Kuznetsova et al., 2018) using CLIP encodings.

**Contrast-Caption**: Inspired by the paradigm of contrastive learning, we reformat existing image captioning tasks into a multi-image version, where 2 tasks are defined. Given multiple images, the first task we defined is to judge which image matches the provided caption, and the second task is to generate a caption for a denoted image. We randomly select 2 to 8 images along with their captions to form a single item, then apply the template of the 2 tasks defined. We used the images and captions provided by ShareGPT-4V (Chen et al., 2023) and LAION GPT-4V captions from LVIS (Schuhmann & Bevan, 2023).

**Spot-the-diff (Jhamtani & Berg-Kirkpatrick, 2018)**: This dataset requires the model to generate the difference description given 2 images. We use the processed version from MIMIC-IT (Li et al., 2023a).

**LRV-multi (Liu et al., 2023a)**: This is a single-image dataset used to mitigate hallucinations of LMMs. We process it similarly to LLaVA-665k-multi to get a "synthetic" multi-image dataset.

**Birds-to-Words (Forbes et al., 2019)**: The task of this dataset is to generate different descriptions and give 2 images of different birds. We directly take the training split for training. Besides, we prompt GPT-3.5-turbo to generate a possible question given the reference answer, so we can get a more diverse question pool instead of always asking the model to generate the difference description. **Multi-VQA**: This dataset contains multi-image reasoning QA pairs, where each question is generated by GPT-4 based on the captions of the images. Each question is required to involve at least two images. Images and captions are sampled from ShareGPT-4V (Chen et al., 2023). The prompt template is in  Table 2.

**Video QA**: We also include some video question answering datasets, including NExTQA (Xiao et al., 2021), STAR (Wu & Yu, 2021), and visual-story-telling (Huang et al., 2016). nextqa and star are the processed version from Flipped-VQA (Ko et al., 2023). The visual-story-telling are the processed version from MIMIC-IT (Li et al., 2023a).

**Single-image-VQA**: Instead of focusing only on multi-image tasks, we claim that it's necessary to also include some single-image VQA dataset to avoid catastrophic forgetting on single-image tasks. We included some of the single image data from LLaVA-665k as well as DocVQA (Mathew et al., 2020), DVQA (Kafle et al., 2018), and ChartQA (Masry et al., 2022) to enhance its ability on diagrams and OCR. In practice, since the size of DVQA is too large, we only use 30k out of 200k examples for the training.

## A.2 Investigation of Mantis's capability on open-ended tasks

To assess Mantis's performance on open-ended generation tasks, we evaluated it on image captioning (using the CoCo-2017-Lite dataset) (Chen et al., 2015) and video captioning (using the Vatex dataset) (Wang et al., 2019). For evaluation, we utilized lmms-eval Li* et al. (2024). The results are presented in Table 9 and Table 10. As shown in these tables, Mantis-Idefics2's captioning performance improved after training on Mantis-Instruct compared to its previous checkpoint Idefics2, though still fall behind of the expert captioning models in these areas.

We attribute this improvement to the design of our dataset, which encourages the model to differentiate between images and better understand semantic information within visual content. Notably, contrastive learning of this kind is often limited in vision-language pre-training datasets, so fine-tuning on Mantis-Instruct appears to enhance the model's comprehension of visual content further.

| Model | BLEU-4 | BLEU-3 | BLEU-2 | BLEU-1 | METEOR | ROUGE-L | CIDEr |
|---|---|---|---|---|---|---|---|
| Idefics2 | 0.1890 | 0.2810 | 0.4128 | 0.5896 | 0.2453 | 0.4545 | 0.7453 |
| Mantis-Idefics2 | 0.2288 | 0.3266 | 0.4570 | 0.6155 | 0.2670 | 0.4906 | 0.8210 |
| Mantis-SIGLIP-LLaMA3 | **0.2474** | **0.3479** | **0.4824** | **0.6500** | **0.2692** | **0.5045** | **0.9042** |

Table 9: Results of Mantis on the image captioning task (CoCo-2017-Lite)

| Model | BLEU-4 | METEOR | ROUGE-L | CIDEr |
|---|---|---|---|---|
| Idefics2 | 0.1454 | 0.1448 | 0.3183 | 0.2307 |
| Mantis-Idefics2 | **0.1702** | **0.1573** | **0.3724** | **0.2667** |
| Mantis-SIGLIP-LLaMA3 | 0.1545 | 0.1431 | 0.3607 | 0.2263 |

Table 10: Results of Mantis on the video captioning task (Vatex)

## A.3 Investigation of Mantis's capability on long-context scenarios

While claiming Mantis to be a model that can accept text-image interleaved inputs, it's interesting to investigate how Mantis will perform in long-context scenarios.

We first conduct experiments to see how Mantis's performance will change as the number of images/frames increases. We run Mantis-Idefics2 on MVBench using 2/4/8/16 frames and report the performance in Table 11. It turns out that Mantis's performance is stable across various numbers of sampled frames. What's worth noting is that Mantis can use only 2 frames to achieve 48.38 accuracy, which means that most of the questions in the MVBench can be answered by keyframes. As the number of frames goes up from 2 to 8, the accuracy also goes up to 51.38 accuracy. The performance under 16 samples is 1 point lower than the performance under 8 frames, we hypothesize that the redundant information might distract Mantes from giving the current answer.

| Number of frames | 2 | 4 | 8 | 16 |
|---|---|---|---|---|
| MVBench Acc | 48.38 | 50.02 | **51.38** | 50.2 |

Table 11: Mantis-Idefics2's performance on MVBench with different number of uniformly sampled frames.

As shown in Table 4, MileBench consists of examples with an average of 15 images per example and a sequence length ranging from 1,000 to 9,000 tokens, making it well-suited for evaluating Mantis's long-context ability. As shown in Table 5, Mantis-SIGLIP achieves 47.5 on MileBench, only 5.5 points behind GPT-4V. We do believe this is good evidence that Mantis maintains competitive multimodal long-context ability.

### A.4 Comparison with MIMIC-IT

MIMIC-IT is also a multi-modal dataset that aims to boost the model's ability for in-context learning, which also contains multiple images for most of the QA pair (Li et al., 2023a). To investigate how well Mantis-Instruct outperforms MIMIC-IT, we conduct a fair comparison between the Otter and Mantis-Flamingo. As shown in Table 12, Otter is tuned from the Open-Flamingo using MIMIC-IT as the instruction-tuning dataset, and one of our Mantis's variants, Mantis-Flamingo, is also tuned from the Open-Flamingo but instead using our Mantis-Instruct as the instruction-tuning dataset. Therefore, Mantis-Flamingo and Otter are a fair comparison where they share the same pre-training dataset, MMC4+LAION, and trained on different instruction-following datasets, MIMIC-IT and Mantis-Instruct, respectively.

We report the performance in Table 13. It turns out that Mantis-Flamingo is better than Otter in all these benchmarks, based on the same pre-trained model but different instruction-tuning datasets. This proves the superior quality of Mantis-Instruct compared to the MIMIC-IT. This is mainly because Mantis-Instruct is designed to equip models with the 4 essential abilities for multi-image reasoning, while MIMIC-IT is mainly designed for multimodal in-context reasoning.

| Model | Pre-training | Instruction-Tuning | Size |
|---|---|---|---|
| Otter | MMC4, LAION | MIMIC-IT | 9B |
| Mantis-Flamingo | MMC4, LAION | Mantis-Instruct | 9B |

Table 12: Comparison of Mantis-Flamingo and Otter's training datasets. Mantis-Flamingo and Otter are a fair comparison where they share the same pre-training dataset, MMC4+LAION, and trained on different instruction-following datasets, MIMIC-IT (Li et al., 2023a) and Mantis-Instruct, respectively

| Model | NVLR2 | Q-Bench | Mantis-Eval | BLINK | MVBench | Average |
|---|---|---|---|---|---|---|
| Otter | 49.15 | 17.50 | 13.29 | 36.26 | 15.30 | 26.30 |
| Mantis-Flamingo | **52.96** | **46.80** | **32.72** | **38.00** | **40.83** | **42.26** |
| Δ over Otter | +3.81 | +29.30 | +19.43 | +1.74 | +25.53 | +15.96 |

Table 13: Comparison of Mantis-Flamingo and Otter's (Li et al., 2023b) performance in multi-image benchmarks. The result numbers are sourced from Table 5.

