# OpenReview forum: "Mantis: Interleaved Multi-Image Instruction Tuning"
_TMLR — Accepted by TMLR_

### Review · Reviewer_Eh2y · 2024-09-13

**Summary Of Contributions:**

The paper presents Mantis, a model family designed to improve large multimodal model (LMM) performance on multi-image visual language tasks via instruction tuning. It claims that Mantis achieves state-of-the-art (SoTA) performance on multiple benchmarks while using significantly smaller datasets compared to models pre-trained on massive datasets, like Idefics. The authors propose Mantis-Instruct, a dataset focused on multi-image tasks, and perform extensive evaluations across both multi-image and single-image benchmarks.

**Audience:**

Yes

**Broader Impact Concerns:**

The broader impacts are well stated and addressed in the paper.

**Claims And Evidence:**

Yes

**Requested Changes:**

Dataset Baselines: Compare the proposed Mantis dataset with existing multi-image instruction tuning datasets on a fixed base model.

Failure Mode Analysis: The addition of qualitative analysis—particularly on tasks where Mantis underperforms—would provide valuable insights for further tuning and refinement of the model.

Longer Output Tasks: Including tasks that require long-form reasoning (rather than shorter multiple-choice answers) would better evaluate the model’s ability to handle complex multi-image scenarios.

Scalability and Efficiency Metrics: Provide more concrete details on how the model performance scales with task complexity (e.g., the number of tasks, the number of images, or sequence length).

Challenges of Dataset Construction: Describe the challenges when making the dataset, and how they were addressed in the study or how they may be addressed in the future.

**Strengths And Weaknesses:**

- Strengths

Clear Motivation: The paper identifies a relevant gap—multi-image capabilities in LMMs—and aims to address it through instruction tuning, rather than relying on extensive pre-training.

Comprehensive Evaluation: Mantis is tested on both multi-image and single-image benchmarks, showing competitive or better performance than other models in the same category. The results are backed by strong empirical data from diverse benchmarks.

Dataset Construction: The creation of Mantis-Instruct, a dataset that systematically targets multi-image skills (comparison, co-reference, reasoning, and temporal understanding), is a well-thought-out contribution. It enhances the ability to test models on specific visual reasoning tasks.

Efficiency Claims: Mantis reportedly achieves superior performance with significantly fewer data and computational resources than Idefics and OpenFlamingo, which is a notable achievement in an era where scaling data is a challenge.

- Weaknesses

Baselines of the Dataset

The idea of multi-image instruction tuning is not necessarily novel.  Though the previous works focus on in-context learning capabilities unlocked by those multi-image instructions, these previous works should be compared as baselines, given the paper claims the dataset as a contribution. An obvious baseline is MIMIC-IT, which the paper borrows data from.

Limited Analysis of Failure Cases

While the paper emphasizes Mantis’s strengths across various benchmarks, it does not provide sufficient analysis of its failure modes. In particular, it would be useful to understand how and why Mantis underperforms in specific tasks. For example, is the degradation in performance on certain single-image benchmarks due to its focus on multi-image tuning? Providing qualitative or failure case analysis could help guide future improvements. Generally, more visual analysis will be appreciated.

Potential Over-Reliance on Specific Skills

The focus on four core skills (co-reference, comparison, reasoning, and temporal understanding) might overlook other important multimodal reasoning abilities. Additional exploration of tasks involving spatial reasoning or higher-level semantics (e.g., abstract reasoning across images) could help further generalize the model.

---

> ### Author Response · Authors · 2024-10-21
> **Author's response to the review**
>
> Dear Reviewer Eh2y,
>
> Thank you for the detailed review and constructive suggestions. We are glad to hear that you think our work has a clear motivation and comprehensive evaluation. For the requested changes, we have responded as follows and hope they can help resolve your concern:
>
> ---
>
> > For the request to include MIMIC-IT
>
> **O1**: MIMIC-IT is a multi-image instruction-following dataset that has been used to train the Otter model, which we have already included as one of our baselines in the paper. We thus think that the comparison between Mantis-Instruct and MIMIC-IT has already been addressed in our experiments.
>
> Additionally, the datasets we borrowed from MIMIC-IT are limited to specific subsets—Spot-the-Diff (8K), NextQA (3K), and STAR (4K)—and the purpose we used the MIMIC-IT’s version is primarily for easier processing. It’s important to note that MIMIC-IT does not claim originality over these datasets.
> Furthermore, the total size of the borrowed datasets (15K) is significantly smaller than both Mantis-Instruct (721K) and MIMIC-IT (2 million), making Mantis-Instruct a largely distinct and novel dataset collection with minimal overlap with MIMIC-IT.
>
> ---
>
> > For the request to include more failure case analysis
>
> **O2**: While Mantis does experience some performance decreases on single-image tasks such as TextVQA and OKVQA, the reduction is within 4%, which we consider acceptable. Our analysis of Mantis-Idefics2 shows that most failure patterns are similar to those of the original Idefics2 before tuning on Mantis-Instruct, including challenges such as lack of knowledge and visual misconceptions. We believe a better view of the slight performance decrease is a reasonable trade-off between multi-image and single-image capabilities. Therefore, the overall performance distribution, rather than individual failure cases, should be the focus.
>
> Additionally, the original Idefics2 had been fine-tuned on nearly all existing vision tasks at the time, potentially reaching an optimal point where further fine-tuning could lead to performance declines due to the forgetting issue. To mitigate this, one potential solution is to replay previously seen datasets to counteract forgetting. We consider this a promising direction for future work.
>
>
> ---
>
> > For the request to include evaluation of a longer output task
>
> **O3**: We have added image captioning (CoCo-2017-Lite), and video captioning task (Vatex) below. Given the limited time, we only run experiments on 2 mantis models and 1 strong baseline model Idefics2.
>
> 1. Image captioning on CoCo-2017-lite
>
> |                      | BLEU-4 | BLEU-3 | BLEU-2 | BLEU-1 | METEOR | ROUGE-L | CIDEr  |
> |----------------------|--------|--------|--------|--------|--------|---------|--------|
> | Idefics2             | 0.1890 | 0.2810 | 0.4128 | 0.5896 | 0.2453 |  0.4545 | 0.7453 |
> | Mantis-Idefics2      | 0.2288 | 0.3266 | 0.4570 | 0.6155 | 0.2670 |  0.4906 | 0.8210 |
> | Mantis-SIGLIP-LLaMA3 | **0.2474** | **0.3479** | **0.4824** | **0.6500** | **0.2692** |  **0.5045** | **0.9042** |
>
>
> 2. Video captioning on Vatex (8 frames, zero-shot):
>
> |                      | BLEU-4 | METEOR | ROUGE-L | CIDEr  |
> |----------------------|--------|--------|---------|--------|
> | Idefics2             | 0.1454 | 0.1448 |  0.3183 | 0.2307 |
> | Mantis-Idefics2      | **0.1702** | **0.1573** |  **0.3724** | **0.2667** |
> | Mantis-SIGLIP-LLaMA3 | 0.1545 | 0.1431 |  0.3607 | 0.2263 |
>
> Clearly, compared to Idefics2, the mantis models still maintain a good (even better) performance on normal image captioning or video caption ability. We will add the full evaluation on all baselines in the final version.

---

> > ### Author Response · Authors · 2024-10-21
> > **Author's response to the review (second part)**
> >
> > ---
> >
> > > For the request to include how the model’s performance scales as the complexity increases
> >
> > **O4**: We have already included 8 diverse multi-image benchmarks, which we believe provide a comprehensive evaluation across tasks of varying difficulty and complexity. For instance, NLVR2 tests logical reasoning, BLINK evaluates low-level vision, MVBench assesses video understanding, and MileBench measures multimodal long-context capabilities. We are confident that these benchmarks are sufficiently complex and thorough to demonstrate the capabilities of the Mantis models.
> >
> > Regarding the evaluation of long-context ability and the number of images, we would like to highlight that MileBench specifically addresses multimodal long-context abilities, with an average of 15 images per example, making it well-suited for evaluating this aspect.
> > In terms of sequence length, Table 4 provides an estimated input sequence length for each benchmark, with ranges from 1,000 to 9,000 tokens, which we believe is broad and comprehensive enough to capture the full spectrum of model performance.
> >
> >
> > ---
> >
> > > For the request to include challenges when making the dataset
> >
> > **O5**: We have provided a detailed breakdown introduction to each subset in the Mantis-Instruct in Appendix A-1. We have included the motivation behind the dataset, the types and formats of the questions, as well as a detailed explanation of how the datasets were constructed. Heuristics are used during the curation is also explained in section 2.3. We think these details have covered the challenges you asked for. Please refer to them for details.
> >
> >  Authors

---

> > ### Comment · Reviewer_Eh2y · 2024-10-23
> > **Concerns Paritally Addressed**
> >
> > Thank you for the updates. I think my words are misinterpreted.
> >
> > I request the authors to benchmark the existing instruction tuning datasets with the Mantis model, i.e., compare finetuning on MIMIC-IT vs. on Mantis-Instruct. I am not asking why Mantis differentiates from MIMIC-IT; it’s well-acknowledged. The point is that the paper claims the dataset as a contribution, but I don’t see a fair comparison that compares Mantis and other datasets on a same pretrained model.

---

> ### Author Response · Authors · 2024-10-25
> **Author's reply about the fair comparison with MIMIC-IT dataset**
>
> Sorry for the misinterpretation and thanks for the follow-up comment. We have checked the Otter paper and confirmed that Otter is tuned from the Open-Flamingo using MIMIC-IT, whereas our Mantis-Flamingo is also tuned from the Open-Flamingo but instead using our Mantis-Instruct. Therefore, **Mantis-Flamingo** and **Otter** are a fair comparison where they share the same pre-training dataset, MMC4+LAION, and trained on different instruction-following datasets, MIMIC-IT and Mantis-Instruct, respectively. We include the detailed comparison results in the following table (*Note that these numbers are all copied by the original paper*):
>
> | Model              | Pre-training | Instruction-Tuning | Size | NVLR2 | Q-Bench | Mantis-Eval | BLINK | MVBench | Average |
> |--------------------|--------------|--------------------|------|-------|---------|-------------|-------|---------|---------|
> | Otter              | MMC4, LAION  | MIMIC-IT           | 9B   | 49.15 |   17.50 |       13.29 | 36.26 |   15.30 |   26.30 |
> | Mantis-Flamingo    | MMC4, LAION  | Mantis-Instruct    | 9B   | **52.96** |   **46.80** |       **32.72** | **38.00** |   **40.83** |   **42.26** |
> | $\Delta$ over Otter | -            | -                  | -    |  3.81 |   29.30 |       19.43 |  1.74 |   25.53 |   15.96 |
>
> **Clearly, it turns out that Mantis-Flamingo is better than Otter in all these benchmarks, based on the same pre-trained model but different instruction-tuning datasets. We believe this proves the superior quality of Mantis-Instruct compared to the MIMIC-IT.** We will also include this comparison in the final version of the paper.

---

### Review · Reviewer_berU · 2024-09-25

**Summary Of Contributions:**

This paper addresses the instruction tuning of large vision-language models (LVLMs) in a multi-image setting, bridging the gap between traditional single-image instruction tuning and real-world applications that require reasoning over multiple images. The authors introduce a new dataset, Mantis-Instruct, specifically designed for this purpose, along with details of their training process. Using this dataset, they fine-tune a family of LVLMs, demonstrating improved performance across several multi-image datasets and benchmarks, including a novel dataset introduced in the paper.

**Audience:**

Yes

**Broader Impact Concerns:**

No ethical concerns.

**Claims And Evidence:**

Yes

**Requested Changes:**

Overall, this paper makes a significant contribution and is worthy of acceptance. The following points, if addressed, could further enhance its impact.
* A detailed discussion of previous and concurrent work on multi-image instruction tuning, such as VILA and LLaVA-OneVision. It would contextualize the contribution and ensure the novelty of the approach is clearly established.
* Expanding the evaluation beyond MCQA to include open-ended tasks would strengthen the work. While not critical for acceptance, this would significantly enhance the generalizability and impact of the paper’s findings.
* Providing more detailed statistics on the Mantis-Instruct dataset would strengthen the paper. While not critical for acceptance, these details would improve the transparency and utility of the dataset for future research.
* A deeper analysis of the influence of interleaving images and text during training could strengthen the work. Although not critical for acceptance, understanding the impact of different interleaving strategies could offer valuable insights for future data curation.

**Strengths And Weaknesses:**

**Strengths**
* *Crucial Research Problem*: The paper addresses an important gap in the field by shifting the focus from single-image instruction tuning to multi-image settings, which better reflects real-world applications that require reasoning over multiple images.
* *Novel Dataset*: The authors introduce Mantis-Instruct, a novel dataset specifically tailored for multi-image instruction tuning. This contribution is significant, as it provides a valuable resource for training and evaluating vision-language models in a multi-image context.
* *Open-weight Models*: The paper releases a family of LVLMs fine-tuned with the proposed dataset, demonstrating the effectiveness of their approach across multiple models. Additionally, the released models provide valuable resources for supporting future research.
* *Strong Empirical Results*: The paper conducts an extensive evaluation across multiple multi-image datasets and tasks, including newly proposed benchmarks, offering strong empirical evidence to validate the effectiveness of their method.

**Weaknesses**
* *Comparison with Concurrent Work*: The paper lacks a comparison or at least a brief discussion of recent concurrent work on multi-image instruction tuning, such as LLAVA-OneVision [1] and LLaVA-NeXT-Interleave [2].
* *Open-ended Evaluation*: The evaluation primarily focuses on multiple-choice question answering (MCQA) tasks, which may not fully capture the model's capability in diverse real-world multi-image scenarios. Broader evaluations on generation tasks like image captioning or open-ended visual question answering could strengthen the findings.
* *Influence of Interleaving Strategy*: The paper emphasize the interleaving format of image and text during training, but a more thorough analysis of its impact on model performance is missing. Exploring how different input formats influence outcomes would provide valuable insights into optimizing training processes.
* *Data Statistics*: The paper mainly provides the detailed data statistics by their source datasets. A more comprehensive breakdown of the Mantis-Instruct dataset, including statistics on the types of images, distribution of questions, complexity, and diversity, would give a better insight into the dataset's representativeness and utility.

[1] Li, Bo, et al. "Llava-onevision: Easy visual task transfer." arXiv preprint arXiv:2408.03326 (2024).

[2] Li, Feng, et al. "Llava-next-interleave: Tackling multi-image, video, and 3d in large multimodal models." arXiv preprint arXiv:2407.07895 (2024).

---

> ### Author Response · Authors · 2024-10-21
> **Author's response to the review**
>
> Dear Reviewer berU,
>
> Thank you for raising concerns and insightful suggestions for our work. We are excited that you think our work is focusing on a crucial research problem, and recognize our novelty in the datasets. For your concerns and requested changes, we have responded to them one by one in the following paragraphs:
>
> ---
>
> > Regarding the comparison with other works
>
> **O1**: **Firstly, we’d like to emphasize that LLaVA-Next-Interleave and LLaVA-OneVision are the follow-up works of Mantis instead of concurrent works.** Mantis was released 3 months prior to these two works, which both have cited our Mantis paper. They even directly incorporated our dataset into their training. Therefore, we believe that these two works are highly influenced by our work and it’s not appropriate to to view them as a concurrent work as Mantis. However, for those works that indeed conducts multi-image tuning for the VLMs, we already discussed and cited them well in either the introduction or the related works, such as Open-Flamingo, Otter, Kosmos-2, Idefics, etc.
>
> ---
>
> > Regarding the lack of open-ended evaluation results
>
> **O2**: We acknowledge the lack of evaluation on open-ended QA questions. The main reason behind this is that we are focusing on evaluating Mantis’s ability of multi-image scenario, and there are not too much open-ended QA designed for multi-image scenario. However, we understand your concern and have now incorporated more results on image captioning (CoCo-2017-Lite), and video captioning task (Vatex) below. Given the limited time, we only run experiments on 2 mantis models and 1 strong baseline model Idefics2.
> Image captioning on CoCo-2017-lite
> |                      | BLEU-4 | BLEU-3 | BLEU-2 | BLEU-1 | METEOR | ROUGE-L | CIDEr  |
> |----------------------|--------|--------|--------|--------|--------|---------|--------|
> | Idefics2             | 0.1890 | 0.2810 | 0.4128 | 0.5896 | 0.2453 |  0.4545 | 0.7453 |
> | Mantis-Idefics2      | 0.2288 | 0.3266 | 0.4570 | 0.6155 | 0.2670 |  0.4906 | 0.8210 |
> | Mantis-SIGLIP-LLaMA3 | **0.2474** | **0.3479** | **0.4824** | **0.6500** | **0.2692** |  **0.5045** | **0.9042** |
>
>
>
> Video captioning on Vatex (8 frames, zero-shot):
> |                      | BLEU-4 | METEOR | ROUGE-L | CIDEr  |
> |----------------------|--------|--------|---------|--------|
> | Idefics2             | 0.1454 | 0.1448 |  0.3183 | 0.2307 |
> | Mantis-Idefics2      | **0.1702** | **0.1573** |  **0.3724** | **0.2667** |
> | Mantis-SIGLIP-LLaMA3 | 0.1545 | 0.1431 |  0.3607 | 0.2263 |
>
> Clearly, compared to Idefics2, the mantis models still maintain a good (even better) performance on normal image captioning or video caption ability. We will add the full evaluation on all baselines in the final version.
>
> ---
>
> > Regarding the influence of the interleaving strategy
>
> **O3**:
> For the image denotation strategies analysis, previous work Co-Instruct (Table 8) [1] has already tried different forms of image denotations like `<img>`, `<img_st><img><img_end>`, `The input image: <img>`, and `The first/second image: <img>`. Their findings demonstrate that explicitly denoting image order, as in `The first/second image: <img>`, yields the best results. Consequently, we adopt this strategy in our work.
> As for the image positioning heuristic, our approach is guided by two key intuitions:
> - (1) Randomly placing images at the beginning or end of the text can enhance the model's generalization capabilities, and
> - (2) When the image plays a functional role in the text, such as in the question "What does <image> imply about the painting style?", it is more appropriate to position the image within the middle of the text.
>
> They are pretty intuitive tricks. Therefore, given the above clarification, we believe it’s okay to omit these ablations in this version of our paper. We will try adding more related experiments about this in the future.
>
>
> ---
>
> > Regarding the dataset details
>
> **We have provided a detailed breakdown introduction to each subset in the Mantis-Instruct in Appendix A-1**. We have included the motivation behind the dataset, the types and formats of the questions, as well as a detailed explanation of how the datasets were constructed. Please refer to this section for details. We also have open-sourced all the construction codes where all the subsets can be re-produced by other users.

---

> > ### Comment · Reviewer_berU · 2024-11-06
> >
> > Given that this is a journal paper and some time has passed since its initial release, I recommend that the authors discuss relevant work, including studies published after the initial release of this paper (but before this submission). This suggestion is not about novelty concerns but rather ensuring completeness. Additionally, note that LLaVA-Next-Interleave was released within one month of this paper, not three months later.

---

### Review · Reviewer_8Nmm · 2024-10-08

**Summary Of Contributions:**

This is a clean, straightforward paper that introduces a new dataset, evaluation benchmark, and pretrained models for the task of visually-conditioned language modeling from *interleaved, multi-image* contexts. The work identifies that existing multimodal large language models (MLLMs) are deficient in their ability to reason over multiple image contexts (e.g., paired images, frames sampled from a video), and identifies four target “capabilities” that a multi-image/interleaved MLLM should have:

1. Coreference: Ability to localize natural language references to the appropriate image in a multi-image context (e.g., “in the second image”). *Nit: I would rename this to be “instance grounding” or “image grounding” — coreference means something very different in NLP)*

2. Comparison: Ability to compare/contrast features *across* multiple images (e.g., spotting the difference between image pairs).

3. Reasoning: Ability to perform “deeper” types of multi-image reasoning across multiple image contexts, such as logical/compositional reasoning, counting, free-form QA.

4. Temporal Understanding: Ability to model the arrow of time, answering questions about ordered frames sampled from videos, still images sampled from comic strips, etc.

To tackle the above, the authors curate and/or gently transform existing datasets into a format amenable to multi-image (interleaved) language modeling. The aggregated dataset — labeled Mantis-Instruct — consists of 721K (text prompt, multi-image context, text response) triples, with a mix of examples that consist of ~5 images on average.

In parallel to the curated Mantis-Instruct data, the authors assemble a multi-image (interleaved) evaluation benchmark consisting of 7 existing benchmarks, along with an 8th newly proposed benchmark — Mantis-Eval — consisting of 217 multi-image examples provided by crowdworkers (using images from Google Search).

To evaluate the impact of the Mantis-Instruct dataset, the authors finetune several pretrained MLLMs (including Fuyu, OpenFlamingo, Idefics2, and two LLaVa variants), in both the full-finetuning and LoRA regime. Through carefully constructed ablation experiments and comparisons to existing (multi-image) MLLMs, the work clearly demonstrates that Mantis-Instruct is a compact and useful dataset for inducing multi-image reasoning abilities.

**Audience:**

Yes

**Broader Impact Concerns:**

The existing Societal Impacts statement does a good job of capturing the general potential for misuse of MLLMs in general. Given that most of the data/evaluations in this work use existing image/text datasets (with proper licenses and attribution), I think most of the ethical concerns I have are addressed.

One thing — the Mantis Eval benchmark created in this work mentions using Google Search to source images. I assume that the authors limited the images to publicly available/creative-commons images — if so, I think adding a sentence about this would be great. If not, then I would want to flag the Mantis-Eval data for additional ethics/licensing review.

**Claims And Evidence:**

Yes

**Requested Changes:**

As it stands the work presents strong empirical contributions, a nice new instruct tuning dataset, and a multi-image evaluation benchmark that has already been seeing wide adoption in recent papers introducing new MLLMs (e.g., LLaVa-NEXT Interleave: https://arxiv.org/abs/2407.07895).

I definitely advocate for accepting this work, but would love for authors to consider the reframing / additional analyses above. Specifically:

1. Understanding the impact of the curation heuristics (importance of image denotations, positioning of <image> token) on downstream performance, as alluded to in Section 2.3.

2. Deeper analysis of how Mantis models use longer multi-image contexts (generalization as a function of K images, relationship to component datasets in Mantis-Instruct)?

3. Analysis of how Mantis models use redundant/video contexts (e.g., what frames do they attend to, what types of MLLM inductive bias is better for longer-context videos).

I would also suggest adding significantly more detail about how Mantis-Eval was collected/constructed; if the authors could provide the interface/process crowdworkers used to produce the 217 examples, that’d be great.

**Strengths And Weaknesses:**

This paper is backed by a large number of carefully controlled finetuning experiments and comparisons across 8 different evaluation tasks (both “held-in” and “held-out” with respect to the Mantis-Instruct data). I specifically want to call attention to the following experiments that I thought were particularly well-executed:

1. “Merge” vs. “Sequence” Input (Sec. 3.4, Table 5) — This is a necessary and often overlooked experiment in the MLLM space that explicitly compares “merging” multi-image contexts into a single image (i.e., as a large resolution mosaic) against explicitly modeling individual images as a sequence. I am very convinced by the results that demonstrate the superior performance of Mantis-tuned MLLMs over strong “single-image” SOTA MLLMs like LLaVa 1.6. Nice work!

2. “Is Multi-Image Pretraining Necessary (Sec 3.6, Table 7) — Like many others in this space, I’ve questioned the value of pretraining on massive datasets like OBELICS or MMC4 for inducing multi-image reasoning in MLLMs; I really appreciate how this set of experiments not only compares Mantis-tuned MLLMs to existing interleaved models like OpenFlamingo and Idefics2, but also systematically ablates multi-image pretraining on OBELICS-100K, and shows no meaningful improvements for downstream tasks. This is a very nice and useful result.

---

However, there are a few areas that I found that paper lacking. First, beyond the construction of the Mantis-Eval benchmark (just 217 examples), there is very little that’s new on the artifact side: all of Mantis-Instruct was compiled from existing datasets (albeit gently reformatted / transformed to suit the instruct tuning format). Similarly, the remaining 7 evaluation benchmarks are similarly compiled from existing work.

Note that is not a problem! I specifically don’t want the authors or other reviewers to think I am penalizing this work for a lack of novelty; rather, I want to call this out explicitly to ask the authors to refocus the framing and details in the paper around the **really strong and interesting experimental results** as well as providing more detail about the individual dataset curation process. Specifically, Section 2.3 mentions a series of heuristics that were seemingly important during dataset curation (e.g., importance of image denotations, positions of the <image> placeholder token) — I would *love* to understand these heuristics and the actual impact of these a lot more.

I also think there’s a lot of space for deeper analysis into the fidelity of multi-image reasoning — how do Mantis-tuned models generalize as the image context grows larger? Is that tied to the relative proportions of (# images / dataset)? When processing redundant image contexts (i.e., frames from a video), do Mantis models need all 8 / 16 / 50 frames, or are the models attending to just a few “keyframes”? I think these analyses are pretty straightforward to do with the existing suite of models, and would really help a lot in solidifying the experimental/empirical contributions of this work.

---

Beyond the above, I found the taxonomy (coreference, comparison, reasoning, temporal understanding) of different “multi-image” reasoning capabilities a bit forced/unmotivated. I understand the desire to categorize each of the different component datasets, but in a lot of cases, it’s not immediately clear that the distinction is meaningful. I think minimally, just splitting into three categories (single-image reasoning, intra-image reasoning, temporal reasoning) might fit a little bit better?

---

> ### Author Response · Authors · 2024-10-21
> **Author's response to the review**
>
> Dear reviewer 8Nmm,
>
> Thanks for the constructive and detailed review, we are glad to see that you think Mantis-Instruct is a “compact and useful dataset for inducing multi-image ability”, and recognized the value of our ablation experiments. Regarding your suggestions for improvements, we hope the following corresponding responses can help address your concerns:
>
> ---
>
> **O1**: We acknowledge the importance of conducting additional ablation studies to evaluate the significance of various heuristics in curating datasets. However, some heuristics have already been validated by previous research, and we believe it is more efficient to leverage those conclusions rather than repeating the experiments, allowing us to allocate computational resources more effectively.
> - For the image denotation, previous work Co-Instruct (Table 8) [1] has already tried different forms of image denotations like `<img>`, `<img_st><img><img_end>`, `The input image: <img>`, and `The first/second image: <img>`. Their findings demonstrate that explicitly denoting image order, as in `The first/second image: <img>`, yields the best results. Consequently, we adopt this strategy in our work.
> - As for the image positioning heuristic, our approach is guided by two key intuitions: (1) Randomly placing images at the beginning or end of the text can enhance the model's generalization capabilities, and (2) When the image plays a functional role in the text, such as in the question "What does <image> imply about the painting style?", it is more appropriate to position the image within the middle of the text. This is a pretty intuitive trick.
>
> Therefore, given the above clarification, we believe it’s okay to omit these ablations in this version of our paper. We will try adding more related experiments about this in the future.
>
> ---
>
> **O2**: **For additional dataset details, we have provided an introduction to each subset of Mantis-Instruct in Appendix A1**. This section covers the motivation behind the dataset, the types and formats of the questions, as well as a detailed explanation of how the datasets were constructed. We encourage you to refer to this section for further information.
>
> ---
>
>
> **O3**: Regarding the investigation of Mantis’s ability in long-context scenarios, as well as how its performance will change as the as number of images/frames increases, we conduct an additional experiment on MVBench to investigate. We sample 2/4/8/16 frames from the MVBench videos and the results of Mantis-Idefics2 are shown below:
>
> |           | MVBench |
> |-----------|---------|
> | 2-frames  |   48.38 |
> | 4-frames  |   50.02 |
> | 8-frames  |   **51.38** |
> | 16-frames |    50.2 |
>
>
> It turns out that Mantis‘s performance is stable across various numbers of sampled frames. What’s worth noting is that Mantis can use only 2 frames to achieve 48.38 accuracy, which means that most of the questions in the MVBench can be answered by keyframes. And as the number of frames goes up from 2 to 8, the accuracy also goes up to 51.38 accuracy. The performance under 16 samples is a 1 point lower than the performance under 8 frames, we hypothesize that the redundant information might distract Mantis’s from giving the current answer.
>
> Besides, some existing multi-image benchmarks also contain QA with various numbers of images and it proves Mantis’s ability on the multimodal long context ability. For example, MileBench has an average 15 number of images per QA example, yet Mantis still presents pretty good performance. We believe that also confirms Mantis’s ability on long-context scenarios.
>
>
> ---
>
> **O4**: Concerning the potential ethical issues related to the Mantis-Eval dataset, we have taken steps to address these concerns before the annotation. All annotators were restricted to using publicly available images, with most of the images sourced from Google Image Search or other public websites. Additionally, we have documented the sources of each curated QA, including relevant links. Based on these precautions, we believe the dataset complies with ethical standards. We will include further clarification on this matter in the final version of our paper.
>
> Authors

---

### Author Response · Authors · 2024-11-10
**Authors' revision on the paper based on reviewer's comments**

We thank all the reviewer for their detailed feedbacks and have updated our paper as follows to address reviewers' requested changes.

1. We have updated details about how we curate different subsets in the Mantis-Instruct in Appendix A.1 to help people understand the additional details behind the curation of Mantis-Instruct. We also have open-sourced all the codes for constructing each subset, which can also be helpful for reproduction. This is requested by reviewers `8Nmm`, `berU`, and `Eh2y`
2. We add investigations on how Mantis performs on open-ended tasks that require long-from outputs in Appendix A.2. Specifically, we evaluate Mantis on image and video captioning tasks and showcase its ability on long-form generation besides the pure MCQ answering. This is requested by reviewers `berU` and `Eh2y`
3. We add experiments about how the performance of Mantis will change in the long context scenario in Appendix A.3. This is requested by `8Nmm`
4. We added experiments about how Mantis's performance will change when the number of frames received changes for video understanding tasks in Appendix A.3. This is requested by reviewer `8Nmm`.
5. We added a detailed comparison with MIMIC-IT, which is also a multimodal instruction-following dataset, by comparing the performance between Otter and Mantis-Flamingo in Appendix A.4. This is requested by reviewer `Eh2y`.
6. We added discussion with contemporary works like LLaVA-Next-Interleave and LLaVA-OneVision in section 4.2 as the recognition of their works. This is requested by reviewer `berU`.
7. We discussed the intuition and reference behind the 3 heuristics used to create the datasets in section 2.3. This is requested by the reviewer `8Nmm`. We also discussed that the image denotation methods come from previous work Co-Instruct, which has conducted detailed comparisons in different multi-image denotation formats, from which we simply select the best one. This is requested by reviewer `berU`

---

### Comment · Editors_In_Chief · 2025-12-02

Congratulation to Mantis on receiving the 2025 Outstanding Certification!

For more details, see https://medium.com/@TmlrOrg/announcing-the-2025-tmlr-outstanding-certification-e26d548ff011.

---

### Decision · Action_Editor_wZFf · 2024-11-13

**Recommendation:** Accept as is

**Comment:**

All reviewers recommended acceptance, based on the reasons summarized under "Claims and Evidence" above. I agree with this consensus.

I thank the authors for already revising their manuscript to address the reviewers' comments, especially for the additional experiments in Appendix A. One minor comment: In the sentence mentioning LLaVA-Next-Interleave and LLaVA-OneVision in Section 4.2, it would be more precise to note that these works appeared after Mantis, which would also explain why these works incorporate Mantis-Instruct.

**Audience:**

It is clear that instilling multi-image capabilities less expensively is of interest, as evidenced by the strengths listed by the reviewers as well as the amount of recent work on the topic both before and after Mantis became available.

**Claims And Evidence:**

The main contribution of this submission is to show that multi-image vision-language modelling capabilities can be achieved by instruction fine-tuning on a dataset, namely Mantis-Instruct, of relatively modest size. All reviewers agree that this claim is supported by strong and comprehensive empirical results on multiple evaluation tasks, both multi-image and single-image. Reviewer 8Nmm calls out two comparisons as being particularly well-executed. In addition, all reviewers valued the contribution of the Mantis-Instruct dataset as well as multiple models fine-tuned on this dataset.